# Association between COVID-19 prognosis and disease presentation, comorbidities and chronic treatment of hospitalized patients

**Alejandro Rodríguez-Molinero**[1]*, **César Gálvez-Barrón**[1], **Antonio Miñarro**[2], **Oscar Macho**[1], **Gabriela F. López**[1], **Maria Teresa Robles**[1], **María Dolores Dapena**[1], **Sergi Martínez**[1], **Núria Milà Ràfols**[1], **Ernesto E. Monaco**[1], **Antonio Hidalgo García**[1], on behalf of the COVID-19 Research Group of CSAPG[¶]

1 Research Area, Consorci Sanitari de l'Alt Penedès i Garraf (CSAPG), Sant Pere de Ribes, Barcelona, Spain, 2 Department of Genetics, Microbiology and Statistics, School of Biology, University of Barcelona, Barcelona, Spain

¶ Membership of the COVID-19 Research Group of CSAPG is provided in the Acknowledgments.
* rodriguez.molinero@gmail.com

## Abstract

### Importance

The rapid pandemic expansion of the disease caused by the new SARS-CoV-2 virus has compromised health systems worldwide. Knowledge of prognostic factors in affected patients can help optimize care.

### Objective

The objective of this study was to analyze the relationship between the prognosis of COVID-19 and the form of presentation of the disease, the previous pathologies of patients and their chronic treatments.

### Design, participants and locations

This was an observational study on a cohort of 418 patients admitted to three regional hospitals in Catalonia (Spain). As primary outcomes, severe disease (need for oxygen therapy via nonrebreather mask or mechanical ventilation) and death were studied. Multivariate binary logistic regression models were performed to study the association between the different factors and the results.

### Results

Advanced age, male sex and obesity were independent markers of poor prognosis. The most frequent presenting symptom was fever, while dyspnea was associated with severe disease and the presence of cough with greater survival. Low oxygen saturation in the emergency room, elevated CRP in the emergency room and initial radiological involvement were all related to worse prognosis. The presence of eosinophilia (% of eosinophils) was an independent marker of less severe disease.

**Data Availability Statement:** Data cannot be shared publicly because of the risk of re-

identification of some patients of the database. Data are available from the Consorci Sanitari de l'Alt Penedès-Garraf (contact: recerca@csapg.cat) for researchers who meet the criteria for access to confidential data.

**Funding:** The authors received no specific funding for this work.

**Competing interests:** The authors have declared that no competing interests exist.

## Conclusions

This study identified the most robust markers of poor prognosis for COVID-19. These results can help to correctly stratify patients at the beginning of hospitalization based on the risk of developing severe disease.

## Introduction

Since the appearance of an outbreak of respiratory disease associated with a new coronavirus (SARS-CoV 2) in Wuhan (China) in December 2019, the spread of this new pathogen in the world population has been continuous, with a pandemic declared on March 11, 2020. Global case fatality rate (about 3,6% of total reported cases in the world) and the total number of affected patients in the world (more than 21 million people on August 16th) makes this new disease (Covid-19) a target of research priority [1].

All health systems in the world are under enormous healthcare pressure due to this pandemic, and Spain has been one of the most affected countries in Europe [1]. In this context, the identification of risk factors or predictors associated with poor prognosis is relevant in terms of early detection of the most vulnerable patients and the best organization of available health resources.

Several studies, including meta-analyses and systematic reviews of cohorts or case series [2–5], have identified various predictors or risk factors for death and severity in patients hospitalized for COVID-19. Thus, several baseline factors (older age and male sex), comorbidities (mainly cardio-vascular pathology), symptoms (dyspnea) and clinical parameters (respiratory function, inflammatory markers and lymphopenia) associated with worse prognosis have been identified. However, the vast majority of these studies come from Asian cohorts, mainly from China. This difference is important because in addition to ethnicity, other determining factors, such as age or associated comorbidity, are quite different. In two reviews of comorbidities in patients with COVID-19 of Asian origin (16 studies, N = 78 520) [6, 7], a relatively low prevalence of hypertension and diabetes mellitus (16–17% and 12–16%, respectively) was reported compared to populations in our environment, such as those analyzed in two Italian cohort studies [8, 9], in which a prevalence of arterial hypertension of 50% and of diabetes mellitus of 17–22% were reported. In Europe, risk factors or predictors have been reported mainly from cohorts of Italy [8–10], the other European country most affected by the pandemic. In Spain, as far as we know, studies of reported risk factors have considered only specific subpopulations, such as renal replacement therapy patients or oncology patients [11–13], or specific laboratory parameters [14].

In the reported cohorts, the association of various chronic pharmacological treatments (with the exception of renin angiotensin-aldosterone blockers) [15–17] with poor prognosis events in COVID-19 patients has not been evaluated. We believe that an exhaustive exploration of this issue is relevant given the high consumption of pharmacological treatments for various chronic pathologies in the countries around us.

Therefore, in this study, we studied the association of various baseline, pharmacological, clinical, radiological and laboratory parameters with adverse clinical events (severe disease and death) in a cohort of patients hospitalized in our health centers.

## Materials and methods

This was an observational cohort study on a sample of 418 patients admitted for COVID-19 to the hospitals of the Consorci Sanitari de l'Alt Penedès i Garraf (CSAPG). The CSAPG is a

consortium of three regional hospitals, serving a total population of 247,357 inhabitants. During the study period, in the reference population served by our hospitals, a total of 1,442 diagnoses of COVID-19 were made by PCR test for SARS-CoV-2 (including community and hospitalized patients). However, this figures does not reflect the incidence of the disease in our area, since PCR test was not performed to patients with mild symptoms, who did not require medical care.

All patients admitted to hour hospitals with a clinical syndrome consistent with COVID-19 were included in the study; those with a negative PCR test for SARS-CoV-2 via nasal smear and those without respiratory involvement were excluded. The data were collected ambispectively, with data collection beginning on April 6, 2020. The data collected corresponded to patients admitted consecutively between the 12ve of March 2020 and the 2nd of May 2020. Information was collected from each patient from the first day of admission until death or discharge.

The data were collected from electronic medical records by the COVID-19 research group of CSAPG, with the help of a digital Case Report Form created in OpenClinica, version 3.1. (Copyright © OpenClinica LLC and collaborators, Waltham, MA, USA). The researchers who collected the data were health care personnel from the center, who received specific training in the data collection procedures. During the data collection process, quality controls were established for the data collected, e.g. checking their consistency and verifying, with the source document, at least 20% of the main variable data. Detected errors were corrected, and when necessary, the responsible researcher was retrained.

Death and severe disease were taken as outcome variables. The latter was defined as the need for oxygen therapy through a nonrebreather mask (approximate inspired fraction of oxygen: 100%) or mechanical ventilation (invasive, noninvasive or high flow nasal cannula).

As exposure variables or risk markers, sex, age and the following blocks of variables were analyzed: (1) previous diseases (comorbidities) and chronic treatments prescribed before admission, (2) data related to the disease presentation of COVID-19 and (3) laboratory analytical parameters at the time of admission.

Previous disease history of the patient was collected dichotomously (Yes/No) after detailed reading of all available patient reports. The list of pathologies recorded in the database included cardiovascular, respiratory, digestive, renal, neoplastic, autoimmune, psychiatric, neurological and other diseases. The complete list of pathologies registered in the database is shown in Table 1.

Chronic treatments prescribed to the patients were also recorded dichotomously (Yes/No) after detailed consultation of the available patient reports and electronic prescriptions. The list of registered drugs included antiplatelet and anticoagulant drugs, analgesics, anti-inflammatories, antidiabetic drugs, drugs for cardiovascular diseases, drugs for the respiratory system, drugs with an effect on the central nervous system, cytotoxic drugs and drugs with action on the immune system, among others. A complete list of registered therapies is also shown in Table 1.

Regarding the disease presentation of COVID-19, the symptoms reported in the emergency reports (dichotomously: cough, fever, dyspnea, anosmia, dysgeusia, diarrhea, arthromyalgia, severe asthenia, skin lesions, headache and confusion), baseline oxygen saturation in the emergency room, affected quadrants on the first chest radiography (range: 0 to 4 quadrants) and C-reactive protein (CRP; mg/L) in the emergency room were recorded.

The following analytical parameters were recorded at admission: PCR results for SARS-CoV-2, hemoglobin, platelets, neutrophils (absolute and percentage), lymphocytes (absolute and percentage), eosinophils, prothrombin time (INR), D-dimer, fibrinogen, glycemia,

**Table 1. Chronic conditions and treatments of hospitalized patients with COVID-19.**

| | Total | Mild D. | Severe D. | OR (95% CI) | p* | Survived | Deceased | OR (95% CI) | p* |
|---|---|---|---|---|---|---|---|---|---|
| | N | n (%) | n (%) | | | n (%) | n (%) | | |
| **Male sex** | 238 | 94 (39.5) | 144 (60.5) | 1.73 (1.17–2.57) | 0.010 | 193 (81.1) | 45 (18.9) | 0.99 (0.61–1.64) | 1.000 |
| **Age** (mean) | 418 | 189 (63.6) | 229 (66.9) | - | 0.180 | 339 (61.9) | 79 (80.4) | - | <0.001 |
| **Chronic kidney disease** | 61 | 20 (32.8) | 41 (67.2) | 1.83 (1.04–3.32) | 0.160 | 34 (55.7) | 27 (44.3) | 4.64 (2.57–8.34) | <0.001 |
| **Hypertension** | 217 | 88 (40.6) | 129 (59.4) | 1.48 (1.00–2.18) | 0.189 | 152 (70.0) | 65 (30.0) | 5.64 (3.13–1087) | <0.001 |
| **Diabetes** | 99 | 35 (35.4) | 64 (64.6) | 1.70 (1.07–2.74) | 0.134 | 66 (66.7) | 33 (33.3) | 2.96 (1.75–4.99) | <0.001 |
| **Dyslipidemia** | 145 | 55 (37.9) | 90 (62.1) | 1.57 (1.05–2.39) | 0.141 | 107 (73.8) | 38 (26.2) | 2.01 (1.22–3.31) | 0.026 |
| **Obesity** | 74 | 23 (31.1) | 51 (68.9) | 2.06 (1.22–3.58) | 0.050 | 59 (79.7) | 15 (20.3) | 1.12 (0.58–2.06) | 0.879 |
| **Smoking** | 36 | 16 (44.4) | 20 (55.6) | 1.03 (0.52–2.10) | 1.000 | 33 (91.7) | 3 (8.3) | 0.38 (0.09–1.11) | 0.228 |
| **Alcoholism** | 11 | 1 (9.1) | 10 (90.9) | 7.59 (1.42–188.85) | 0.089 | 10 (90.9) | 1 (9.1) | 0.48 (0.02–2.57) | 0.840 |
| **Heart failure** | 26 | 9 (34.6) | 17 (65.4) | 1.59 (0.70–3.85) | 0.604 | 13 (50.0) | 14 (53.8) | 5.82 (2.55–13.49) | <0.001 |
| **Ischemic heart disease** | 37 | 18 (48.6) | 19 (51.4) | 0.86 (0.43–1.71) | 1.000 | 80 (216.2) | 7 (18.9) | 1.02 (0.39–2.30) | 1.000 |
| **Aortic valve disease** | 10 | 5 (50.0) | 5 (50.0) | 0.82 (0.22–3.10) | 1.000 | 2 (20.0) | 8 (80.0) | 17.81 (4.24–131.51) | <0.001 |
| **Mitral valve disease** | 11 | 9 (27.3) | 8 (72.7) | 2.17 (0.60–10.57) | 0.652 | 6 (54.5) | 5 (45.5) | 3.75 (1.02–13.13) | 0.091 |
| **Pulm. valve disease** | 2 | 1 (50.0) | 1 (50.0) | 0.82 (0.02–32.32) | 1.000 | 2 (100.0) | 0 (0.0) | - | 1.000 |
| **Pacemaker** | 6 | 3 (50.0) | 3 (50.0) | 0.82 (0.14–4.84) | 1.000 | 1 (16.7) | 5 (83.3) | 20.38 (3.07–544.24) | 0.004 |
| **Other heart disease** | 9 | 2 (22.2) | 7 (77.8) | 2.79 (0.65–20.89) | 0.460 | 4 (44.4) | 5 (55.6) | 5.58 (1.39–24.06) | 0.040 |
| **Atrial fibrillation** | 45 | 16 (35.6) | 29 (64.4) | 1.56 (0.83–3.04) | 0.477 | 23 (51.1) | 22 (48.9) | 5.27 (2.74–10.16) | <0.001 |
| **Stroke** | 23 | 6 (26.1) | 17 (73.9) | 2.40 (0.97–6.88) | 0.248 | 13 (56.5) | 10 (43.5) | 3.63 (1.48–8.67) | 0.016 |
| **Gastropathy** | 32 | 13 (40.6) | 19 (59.4) | 1.22 (0.59–2.61) | 1.000 | 23 (71.9) | 9 (28.1) | 1.78 (0.75–3.92) | 0.283 |
| **Inflam. bowel disease** | 5 | 3 (60.0) | 2 (40.0) | 0.56 (0.06–3.71) | 0.955 | 4 (80.0) | 1 (20.0) | 1.18 (0.04–8.68) | 1.000 |
| **Celiac disease** | 3 | 1 (33.3) | 2 (66.7) | 1.56 (0.13–49.10) | 1.000 | 3 (100.0) | 0 (0.0) | - | 1.000 |
| **Chronic hepatitis C** | 0 | 0 | 0 | - | | 0 | 0 | - | - |
| **Other liver disease** | 24 | 7 (29.2) | 17 (70.8) | 2.06 (0.86–5.49) | 0.364 | 17 (70.8) | 7 (29.2) | 1.86 (0.69–4.53) | 0.314 |
| **Arthritis** | 1 | 0 (0.0) | 1 (100.0) | - | 1.000 | 1 (100.0) | 0 (0.0) | - | 1.000 |
| **Spondyloarthritis** | 2 | 1 (50.0) | 1 (50.0) | 0.82 (0.02–32.32) | 1.000 | 2 (100.0) | 0 (0.0) | - | 1.000 |
| **Other autoimmune** | 18 | 4 (22.2) | 14 (77.8) | 2.92 (1.02–10.77) | 0.189 | 10 (55.6) | 8 (44.4) | 3.70 (1.35–9.85) | 0.030 |
| **Asthma** | 23 | 11 (47.8) | 12 (52.2) | 0.89 (0.38–2.13) | 1.000 | 21 (91.3) | 2 (8.7) | 0.42 (0.06–1.49) | 0.434 |
| **COPD** | 41 | 14 (34.1) | 27 (65.9) | 1.66 (0.85–3.37) | 0.364 | 29 (70.7) | 12 (29.3) | 1.92 (0.90–3.90) | 0.183 |
| **OSAS** | 34 | 11 (32.4) | 23 (67.6) | 1.79 (0.86–3.94) | 0.372 | 22 (64.7) | 12 (35.3) | 2.59 (1.18–5.43) | 0.051 |
| **Pulmonary hypert.** | 3 | 1 (33.3) | 2 (66.7) | 1.56 (0.13–49.10) | 1.000 | 2 (66.7) | 1 (33.3) | 2.29 (0.07–28.64) | 0.644 |
| **Other lung disease** | 18 | 7 (38.9) | 11 (61.1) | 1.30 (0.50–3.66) | 0.939 | 16 (88.9) | 2 (11.1) | 0.56 (0.08–2.04) | 0.727 |
| **Depression** | 63 | 29 (46.0) | 34 (54.0) | 0.96 (0.56–1.66) | 1.000 | 45 (71.4) | 18 (28.6) | 1.93 (1.02–3.53) | 0.115 |
| **Schizophrenia** | 4 | 2 (50.0) | 2 (50.0) | 0.82 (0.09–7.98) | 1.000 | 2 (50.0) | 2 (50.0) | 4.36 (0.45–42.37) | 0.283 |
| **Other psych. dis.** | 29 | 13 (44.8) | 16 (55.2) | 1.01 (0.47–2.22) | 1.000 | 22 (75.9) | 7 (24.1) | 0.42 (0.54–3.32) | 0.644 |
| **Dementia** | 43 | 19 (44.2) | 24 (55.8) | 1.05 (0.55–2.00) | 1.000 | 19 (44.2) | 24 (55.8) | 7.28 (3.74–14.40) | <0.001 |
| **Parkinson's disease** | 2 | 1 (50.0) | 1 (50.0) | 0.82 (0.02–32.32) | 1.000 | 1 (50.0) | 1 (50.0) | 4.31 (0.11–169.14) | 0.512 |
| **Multiple sclerosis** | 2 | 1 (50.0) | 1 (50.0) | 0.82 (0.02–32.32) | 1.000 | 1 (50.0) | 1 (50.0) | 4.31 (0.11–169.14) | 0.512 |
| **Other neurodeg. dis.** | 9 | 3 (33.3) | 6 (66.7) | 1.63 (0.41–8.25) | 0.833 | 5 (55.6) | 4 (44.4) | 3.57 (0.83–14.33) | 0.143 |
| **Lung Ca** | 4 | 0 (0.0) | 4 (100.0) | - | 0.351 | 2 (50.0) | 2 (50.0) | 4.36 (0.45–42.37) | 0.283 |
| **Breast Ca** | 7 | 5 (71.4) | 2 (28.6) | 0.34 (0.04–1.67) | 0.531 | 6 (85.7) | 1 (14.3) | 0.79 (0.03–4.91) | 1.000 |
| **Hepatocell. carcinoma** | 3 | 1 (33.3) | 2 (66.7) | 1.56 (0.13–49.10) | 1.000 | 2 (66.7) | 1 (33.3) | 2.29 (0.07–28.64) | 0.644 |
| **Other digestive Ca** | 7 | 3 (42.9) | 4 (57.1) | 1.09 (0.23–5.97) | 1.000 | 5 (71.4) | 2 (28.6) | 1.81 (0.23–8.96) | 0.786 |
| **Other cancer** | 25 | 11 (44.0) | 14 (56.0) | 1.05 (0.476–2.44) | 1.000 | 19 (76.0) | 6 (24.0) | 1.41 (0.49–3.49) | 0.771 |
| **Hematologic neoplasia** | 2 | 1 (50.0) | 1 (50.0) | 0.82 (0.02–32.32) | 1.000 | 1 (50.0) | 1 (50.0) | 4.31 (0.11–169.14) | 0.512 |
| **HIV** | 3 | 0 (0.0) | 3 (100.0) | - | 0.531 | 2 (66.7) | 1 (33.3) | 2.29 (0.07–28.64) | 0.644 |
| **Organ transplant** | 1 | 0 (0.0) | 1 (100.0) | - | 1.000 | 1 (100.0) | 0 (0.0) | - | 1.000 |

*(Continued)*

**Table 1.** (Continued)

| | Total | Mild D. | Severe D. | OR (95% CI) | p* | Survived | Deceased | OR (95% CI) | p* |
|---|---|---|---|---|---|---|---|---|---|
| | N | n (%) | n (%) | | | n (%) | n (%) | | |
| Other immunosupr. | 5 | 3 (60.0) | 2 (40.0) | 0.56 (0.06–3.71) | 0.954 | 5 (100.0) | 0 (0.0) | - | 0.768 |
| Thyroid disease | 31 | 15 (48.4) | 16 (51.6) | 0.87 (0.41–1.84) | 1.000 | 27 (87.1) | 4 (12.9) | 0.64 (0.18–1.70) | 0.653 |
| Anemia | 33 | 12 (36.4) | 21 (63.6) | 1.50 (0.71–3.20) | 0.652 | 21 (63.6) | 12 (36.4) | 2.71 (1.23–5.75) | 0.047 |
| Blood dis. not cancer | 6 | 4 (66.7) | 2 (33.3) | 0.42 (0.05–2.33) | 0.717 | 5 (83.3) | 1 (16.7) | 0.95 (0.04–6.29) | 1.000 |
| Psoriasis | 3 | 2 (66.7) | 1 (33.3) | 0.44 (0.01–5.44) | 0.906 | 2 (66.7) | 1 (33.3) | 2.29 (0.07–28.64) | 0.644 |
| Paracetamol | 100 | 53 (53.0) | 47 (47.0) | 0.66 (0.42–1.04) | 0.248 | 74 (74.0) | 26 (26.0) | 1.76 (1.02–2.99) | 0.094 |
| NSAIDs | 33 | 17 (51.5) | 16 (48.5) | 0.76 (0.37–1.56) | 0.768 | 26 (78.8) | 7 (21.2) | 1.19 (0.45–2.72) | 0.815 |
| Opioids | 29 | 11 (37.9) | 18 (62.1) | 1.37 (0.64–3.10) | 0.747 | 21 (72.4) | 8 (27.6) | 1.72 (0.69–3.94) | 0.366 |
| Corticosteroids | 19 | 4 (21.1) | 15 (78.9) | 3.15 (1.11–11.51) | 0.151 | 12 (63.2) | 7 (36.8) | 2.66 (0.95–6.95) | 0.136 |
| Antihistamines | 18 | 9 (50.0) | 9 (50.0) | 0.81 (0.31–2.17) | 1.000 | 14 (77.8) | 4 (22.2) | 1.27 (0.34–3.70) | 0.886 |
| Antacids | 130 | 51 (39.2) | 79 (60.8) | 1.42 (0.94–2.18) | 0.307 | 92 (70.8) | 38 (29.2) | 2.48 (1.50–4.11) | 0.002 |
| Insulin | 31 | 13 (41.9) | 18 (58.1) | 1.15 (0.55–2.48) | 1.000 | 22 (71.0) | 9 (29.0) | 1.87 (0.78–4.13) | 0.277 |
| Metformin | 58 | 19 (32.8) | 39 (67.2) | 1.83 (1.03–3.35) | 0.186 | 40 (69.0) | 18 (31.0) | 2.21 (1.16–4.08) | 0.047 |
| Antidiabetics | 38 | 14 (36.8) | 24 (63.2) | 1.46 (0.74–2.98) | 0.604 | 27 (71.1) | 11 (28.9) | 1.88 (0.85–3.90) | 0.239 |
| Lipid-lowering drugs | 100 | 39 (39.0) | 61 (61.0) | 1.39 (0.88–2.22) | 0.408 | 77 (77.0) | 23 (23.0) | 1.40 (0.80–2.40) | 0.386 |
| Inhaled ipratropium | 37 | 11 (29.7) | 26 (70.3) | 2.05 (1.01–4.47) | 0.195 | 28 (75.7) | 9 (24.3) | 1.44 (0.61–3.10) | 0.556 |
| Inhaled beta-2 | 53 | 16 (30.2) | 37 (69.8) | 2.07 (1.13–3.96) | 0.134 | 43 (81.1) | 10 (18.9) | 1.01 (0.46–2.04) | 1.000 |
| Inhaled corticosteroid | 47 | 15 (31.9) | 32 (68.1) | 1.87 (0.99–3.68) | 0.202 | 37 (78.7) | 10 (21.3) | 1.19 (0.54–2.44) | 0.840 |
| Other inhalers | 6 | 3 (50.0) | 3 (50.0) | 0.82 (0.14–484) | 1.000 | 4 (66.7) | 2 (33.3) | 2.25 (0.27–12.45) | 0.492 |
| Antiplatelet agents | 78 | 30 (38.5) | 48 (61.5) | 1.40 (0.85–2.34) | 0.477 | 52 (66.7) | 26 (33.3) | 2.70 (1.54–4.70) | 0.003 |
| Anticoagulants | 34 | 15 (44.1) | 19 (55.9) | 1.05 (0.52–2.16) | 1.000 | 19 (55.9) | 15 (44.1) | 3.94 (1.87–8.19) | 0.002 |
| Diuretics | 103 | 43 (41.7) | 60 (58.3) | 1.20 (0.77–1.90) | 0.726 | 71 (68.9) | 32 (31.1) | 2.57 (1.52–4.31) | 0.002 |
| Antihypertensives | 74 | 29 (39.2) | 45 (60.8) | 1.35 (0.81–2.27) | 0.604 | 54 (73.0) | 20 (27.0) | 1.79 (0.98–3.19) | 0.143 |
| Beta-blockers | 60 | 22 (36.7) | 38 (63.3) | 1.41 (0.80–2.53) | 0.531 | 47 (78.3) | 11 (18.3) | 1.01 (0.48–2.00) | 1.000 |
| ACE inhibitors | 93 | 35 (37.6) | 58 (62.4) | 1.49 (0.93–2.41) | 0.281 | 71 (76.3) | 22 (23.7) | 1.46 (0.82–2.53) | 0.369 |
| ARA-2 | 56 | 20 (35.7) | 36 (64.3) | 1.57 (0.88–2.87) | 0.372 | 41 (73.2) | 15 (26.8) | 1.71 (0.87–3.23) | 0.260 |
| Antiarrhythmics | 15 | 2 (13.3) | 13 (86.7) | 5.27 (1.41–37.09) | 0.089 | 7 (46.7) | 8 (53.3) | 5.30 (1.81–15.87) | 0.009 |
| Sedatives | 87 | 31 (35.6) | 56 (64.4) | 1.64 (1.01–2.73) | 0.189 | 62 (71.3) | 25 (28.7) | 2.07 (1.18–3.56) | 0.038 |
| Antidepressants | 90 | 37 (41.1) | 53 (58.9) | 1.24 (0.77–1.99) | 0.706 | 61 (67.8) | 29 (32.2) | 2.64 (1.53–4.50) | 0.003 |
| Antipsychotics | 42 | 13 (31.0) | 29 (69.0) | 1.95 (0.10–4.00) | 0.189 | 16 (38.1) | 26 (61.9) | 9.78 (4.95–19.90) | <0.001 |
| Antiepileptics | 14 | 6 (42.9) | 8 (57.1) | 1.10 (0.37–3.46) | 1.000 | 9 (64.3) | 5 (35.7) | 2.50 (0.73–7.60) | 0.277 |
| Anti-parkinsonians | 4 | 2 (50.0) | 2 (50.0) | 0.82 (0.09–7.98) | 1.000 | 1 (25.0) | 3 (75.0) | 12.16 (1.39–351.81) | 0.057 |
| Other- SNC | 33 | 13 (39.4) | 20 (60.6) | 1.29 (0.63–2.74) | 0.906 | 22 (66.7) | 11 (33.3) | 2.34 (1.04–4.99) | 0.088 |
| Chemotherapy | 4 | 1 (25.0) | 3 (75.0) | 2.29 (0.26–66.03) | 0.939 | 3 (75.0) | 1 (25.0) | 1.56 (0.05–13.63) | 0.750 |
| Immunotherapy | 13 | 5 (38.5) | 8 (61.5) | 1.32 (0.42–4.54) | 1.000 | 10 (76.9) | 3 (23.1) | 1.34 (0.28–4.60) | 0.857 |

*p value is corrected for multiple comparisons. CNS: Central nervous system. OSAS: Obstructive sleep apnea syndrome.

sodium, creatinine, urea, glomerular filtration, transaminases, bilirubin, LDH, CRP, ferritin, lactate and gasometry parameters.

No *a priori* calculation of the sample size was made because the intention of the researchers was to include the total number of patients available during the study period.

In the statistical analysis, the association of each factor collected with the outcomes of interest (serious illness or death) was explored. First, bivariate comparisons were conducted for each factor with the outcomes, and statistical significance was adjusted according to the high number of comparisons by using the False Discovery Rate technic [18]. Second, multivariate

binary logistic regression models were performed with the most relevant factors of each block of variables, to establish which of the factors were the most robust independent predictors of death or serious disease. In the multivariate models, both variables with statistical association with the outcome, as identified in the bivariate models, and variables of clinical relevance in the opinion of the group of researchers were introduced. Features with less than 15 cases in the sample, were not included in the multivariable models. The variables finally included in the model were preselected using the Lasso method [19], this method helps to control multicollinearity problems, which may arise in models with a large number of variables [20]. The laboratory parameters underwent a logarithmic transformation, in order to improve their adjustment to normality, and also they were scaled, to obtain dimensionless variables of zero mean and standard deviation 1, which would allow Odds Ratio (OR) comparisons between them. Based on the results, some analyses were repeated in the subgroup of patients younger than 80 years to mitigate the important effect of age on prognosis, in part due to limited access to intensive care units, which during the epidemic wave were treating the oldest patients in Spain.

Missing data were only imputed in the case of laboratory values at admission. When results of analyses on day one of admission were not available, results of analyses for the second day were used if available. In this study of prognostic markers, results from analyses performed beyond the first 48 hours of admission were not included. No other missing data were imputed.

The authors confirm that all methods were carried out in accordance with relevant guidelines and regulations, including the Declaration of Helsinki in its latest version and Regulation (EU) 2016/679 of the European Parliament and of the Council of April 27, 2016 on Data Protection (RGPD) and other concordant rules. The research ethics committee of the Hospital de Bellvitge reviewed the study and accepted the waiver of each patient's informed consent, as this study was an observational and ambispective review of clinical data, and each patient's personal data were anonymized for publication.

## Results

Of the 464 patients admitted with clinical suspicion of COVID-19 in the study period, 46 patients were not included in the analysis for having a negative PCR for SARS-CoV-2 (nasal smear) or not having respiratory involvement. Thus, 418 patients were included in the analysis. The mean age of the sample was 65.4 years (SD 16.6 years), and 43.1% were women. The median follow-up was 9.5 days (IQR 7 days). All patients were followed until discharge or until day 30 of admission; therefore, there were no cases censored on the final date of the study. In total, 79 patients died (18.9%, 95% CI 15.1–22.7%), 25 patients were intubated (6.0%, 95% CI 3.7–8.3%) and 229 patients required oxygen therapy via a nonrebreather mask or mechanical ventilation (54.8% 95% CI: 50.0–59.6%).

### Comorbidities and chronic treatment

The different comorbidities that patients presented as well as the chronic treatment they received before contracting COVID-19 are shown in Table 1. The same table shows the odds ratio for death or for developing severe disease associated with each of these factors, as well as the statistical significance corrected by multiple comparisons (bivariate analysis).

In the multivariate models, male sex and obesity were the risk markers most strongly associated with severe disease (need for a nonrebreather mask or mechanical ventilation). In the total sample, age was the only factor independently associated with death, according to the multivariate analysis, adjusted for the other relevant factors (Table 2). When the analysis was

**Table 2. Final multivariable models.**

| Chronic pathologies model | | Disease severity | | | Case fatality | |
|---|---|---|---|---|---|---|
| | **Estimator** | **Odds Ratio** | **p** | **Estimator** | **Odds Ratio** | **p** |
| Age | 0.01 | 1.01 (0.10–1.02) | 0.224 | 0.08 | 1.08 (1.05–1.12) | <0.001 |
| Sex (female) | -0.63 | 0.53 (0.35–0.80) | 0.002 | - | - | - |
| Diabetes Mellitus | 0.28 | 1.32 (0.79–2.21) | 0.293 | 0.54 | 1.71 (0.90–3.26) | 0.100 |
| Dyslipidemia | 0.16 | 1.18 (0.74–1.87) | 0.492 | - | - | - |
| Obesity | 0.74 | 0.09 (0.19–3.66) | 0.010 | - | - | - |
| Chronic kidney disease | 0.43 | 1.154 (0.82–2.88) | 0.177 | 0.41 | 1.51 (0.75–3.04) | 0.250 |
| Hypertension | - | - | - | 0.47 | 1.59 (0.74–3.43) | 0.233 |
| Heart failure | - | - | - | 0.15 | 1.16 (0.44–3.06) | 0.768 |
| Atrial fibrillation | - | - | - | 0.62 | 1.86 (0.86–4.02) | 0.113 |
| Dementia | - | - | - | 0.79 | 2.20 (0.99–4.85) | 0.052 |
| OSAS | - | - | - | 0.75 | 2.11 (0.77–5.73) | 0.145 |
| Auto-inmune disease | - | - | - | 0.82 | 2.28 (0.73–7.08) | 0.156 |
| **Chronic medications model** | | Disease severity | | | Case fatality | |
| | **Estimator** | **Odds Ratio** | **p** | **Estimator** | **Odds Ratio** | **p** |
| Age | 0.01 | 1.01 (0.99–1.02) | 0.080 | 0.09 | 1.10 (1.07–1.13) | <0.001 |
| Sex (female) | -0.64 | 0.53 (0.35–0.80) | 0.003 | -0.64 | 0.53 (0.28–1.01) | 0.052 |
| Obesity | 0.77 | 2.17 (1.24–3.79) | 0.007 | - | - | - |
| Corticosteroids | 1.23 | 3.41 (1.08–10.71) | 0.036 | - | - | - |
| Metformin | 0.47 | 1.61 (0.87–2.96) | 0.130 | - | - | - |
| Inhaled beta-2 | 0.47 | 1.60 (0.83–3.06) | 0.158 | - | - | - |
| Anticoagulants | - | - | - | 0.52 | 1.69 (0.73–3.88) | 0.221 |
| Antipsychotics | - | - | - | 1.74 | 5.69 (2.52–12.85) | <0.001 |

OSAS: Obstructive sleep apnea syndrome.

repeated in the subsample of patients younger than 80 years, the only factor that independently explained case fatality remained age (OR 1.07 for each year added; 95% CI: 1.01–1.12). In multivariate analyses of the set of chronic treatments prescribed to the participants, which were also adjusted by age, sex and obesity, corticosteroids (prescribed before contracting the disease) were an independent predictor of severe disease, and antipsychotics ended up, in the final as predictors of case fatality (Table 2). To further investigate the effect of corticoids, they were introduced into a multivariate model of case fatality, adjusted for chronic pathologies (other than obesity, chronic kidney disease, diabetes and dyslipidemia, were preselected by Lasso method). In this model, corticosteroids continued to present as an independent risk factor (OR 3.47 95% CI: 1.09–11.03). Likewise, to rule out that confounding factors prevented recognizing the risk that we *a priori* assumed associated with ACE inhibitors, these drugs were introduced into a multivariate model of case fatality, adjusted for chronic diseases, which did not show that ACE inhibitors were a risk factor, independent of death or serious illness.

When these analyses were repeated in the subsample of patients younger than 80 years, no treatment was found to be an independent predictor of severe disease or case fatality.

## Disease presentation

The presenting symptoms most frequently reported in histories provided in the emergency room were, in this order, fever (83.0%), cough (68.9%), dyspnea (59.6%), diarrhea (27.8%), asthenia (20.1%), arthromyalgia (17.9%), headache (8.4%), dysgeusia (6.2%), anosmia (5.5%)

and confusion (4.5%). Dyspnea was an important predictor of severe disease (OR 2.71, 95% CI 1.82–4.07), and confusion was an important predictor of death (OR 5.27 95% CI 2.03–13.93). Fewer patients died whose reports reported diarrhea (OR 0.32 95% CI 0.15–0.63), arthromyalgia (OR 0.15 95% CI 0.04–0.43), headache (OR 0.26 95% CI 0.04–0.88) and alterations of smell and taste (none of the 26 patients with smell and taste changes died; p<0.01). The presence of asthenia was associated, on the other hand, with a lower risk of serious disease (OR 0.58 95% CI 0.36–0.95). Notably, cough was strongly associated with a good prognosis (OR 0.16 95% CI 0.09–0.26), as patients with cough died much less frequently (9.4%) than those in whom this symptom was not included in the emergency room reports (40.0%). To rule out that this result was due to the action of age (elderly patients who are at risk of death, typically cough less), age and cough were jointly entered into a multivariate predictive model of death. Both factors turned out to be independent predictors (OR for cough in this model was 0.30; IC95% 0.17–0.55). In addition, the protective role of cough remained in the less than 80 years old sample.

Strong baseline predictors for both severe disease and death were low baseline oxygen saturation in the emergency department (means difference: 5.9% for severe disease and 8.1% for death), high CRP in the emergency room analysis (means difference: 57 mg/L for severe disease, 63.1 mg/L for death) and the number of quadrants affected on chest radiography (means difference: 0.7 quadrants for severe disease 0.6 quadrants for death). The above associations were statistically significant with p value <0.001.

The mean time from symptom onset to emergency care was significantly longer in patients who overcame the disease (8.0 days; SD 4.5) than in those who ended up dying (6.2 days; SD 4.7; p = 0.002). This effect was less marked in the subgroup of patients younger than 80 years (time to emergency room care of the deceased: 6.5 days; SD 4.2; p = 0.053).

### Laboratory analytical parameters

Patients admitted for COVID-19 presented leukocytosis with neutrophilia, eosinophilopenia and lymphopenia. In addition, they presented elevated LDH and acute phase reactants (CRP and ferritin), alterations in coagulation parameters (INR, fibrinogen, D-dimer), renal failure and alterations in transaminases. The differences in these parameters between patients with and without severe disease as well as between deceased patients and survivors can be seen in Table 3.

Multivariate models with different analytical parameters (logarithmic transformed and scaled variables were used) showed that in the total sample, CRP was the best predictor of severe disease (OR 2.33 95% CI 1.71–3.19) and eosinophilia (% of eosinophils) was an independent protective factor (OR 0.67 95% CI 0.50–0.89). The predictive capacity of both parameters remained independent when age and basal oxygen saturation was added to the model, along with analytical parameters.

The risk of death was independently related to increased sodium levels (OR 2.24; IC95% 1.46–3.43), glucose levels (OR 1.62; IC95% 1.15–2.28), urea levels (OR 2.51; IC95% 1.61–3.90) and decreased hemoglobin levels (OR 0.70; IC95% 0.52–0.95). When age and oxygen saturation were added as co-variables, along with laboratory tests, only increased sodium levels remained independently associated with death, along with age.

When these models were repeated in patients younger than 80 years, no analytical parameter of those studied was an independent risk marker of death, although CRP remained independent predictor of serious disease (OR 2.92; IC95% 1.80–4.74).

### Discussion

Among the baseline factors associated with poor prognosis, obesity stands out as the specific parameter of cardiovascular risk that is robustly associated with poor prognosis, being a better

**Table 3.**

| | N | Total Mean (SD) | n | Mild disease Mean (SD) | n | Severe disease Mean (SD) | p | n | Survived Mean (SD) | n | Deceased Mean (SD) | p |
|---|---|---|---|---|---|---|---|---|---|---|---|---|
| Hemoglobin (g/L) | 341 | 13,3 (1,9) | 157 | 13,4 (1,8) | 184 | 13,3 (2) | 1,000 | 270 | 13,5 (1,8) | 71,0 | 12,8 (2,2) | 0,013 |
| Platelets (10e9/L) | 341 | 223,1 (96,0) | 157 | 226,2 (96,3) | 184 | 220,4 (96,0) | 0,630 | 270 | 223,8 (96,0) | 71,0 | 220,6 (96,9) | 0,724 |
| Neutrophils (10e9/L) | 341 | 6 (3,7) | 157 | 5,2 (3,2) | 184 | 6,7 (4,1) | 0,006 | 270 | 5,5 (3,3) | 71,0 | 7,8 (4,6) | <0,001 |
| Neutrophils (%) | 341 | 75,8 (11,8) | 157 | 72,4 (11,0) | 184 | 78,6 (11,8) | 0,006 | 270 | 74,6 (11,1) | 71,0 | 80,3 (13,3) | <0,001 |
| Lymphocytes (10e9/L) | 341 | 1,1 (0,7) | 157 | 1,2 (0,8) | 184 | 1 (0,5) | 0,001 | 270 | 1,1 (0,7) | 71,0 | 1 (0,7) | 0,069 |
| Lymphocytes (%) | 341 | 16,6 (9,5) | 157 | 19,1 (9,4) | 184 | 14,4 (9,0) | 0,001 | 270 | 17,6 (9,1) | 71,0 | 12,8 (10,1) | 0,069 |
| Eosinophils (%) | 341 | 0,3 (0,6) | 157 | 0,5 (0,8) | 184 | 0,2 (0,5) | <0,001 | 270 | 0,4 (0,7) | 71,0 | 0,2 (0,4) | 0,038 |
| Prothrombin (INR) | 334 | 1,2 (0,6) | 154 | 1,1 (0,5) | 180 | 1,2 (0,7) | 0,195 | 263 | 1,1 (0,5) | 71,0 | 1,4 (0,8) | <0,001 |
| D-dimer (ng/ml) | 250 | 1875,2 (2719,3) | 127 | 1461,3 (2266,8) | 123 | 2299,4 (3070,5) | <0,001 | 200 | 1436,9 (2071,1) | 50,0 | 3628,6 (4029,3) | <0,001 |
| Glucose (mg/dL) | 337 | 132,3 (55,9) | 154 | 119,6 (40,8) | 183 | 143,1 (64,1) | <0,001 | 266 | 125,1 (51,3) | 71,0 | 159,4 (63,8) | <0,001 |
| Sodium (mEq/L) | 342 | 139 (5,3) | 156 | 139,1 (5,0) | 186 | 138,9 (5,6) | 1,000 | 270 | 137,8 (3,5) | 72,0 | 143,6 (8,0) | <0,001 |
| Creatinine (mg/dL) | 342 | 1,2 (0,7) | 157 | 1,1 (0,7) | 185 | 1,3 (0,8) | 0,004 | 271 | 1,0 (0,5) | 71,0 | 1,7 (1,1) | <0,001 |
| Urea (mg/dL) | 337 | 48 (40,5) | 155 | 43,7 (41,5) | 182 | 51,7 (39,3) | 0,047 | 265 | 37,4 (24,8) | 72,0 | 87,2 (59,1) | <0,001 |
| Alkaline phosphatase (UI/L) | 241 | 82,6 (66,6) | 119 | 77,9 (52,0) | 122 | 87,2 (78,2) | 0,869 | 206 | 83,4 (70,9) | 35,0 | 77,5 (32,5) | 1,000 |
| AST (UI/L) | 231 | 68,5 (241,8) | 122 | 73,3 (328,7) | 109 | 63,2 (58,9) | 0,041 | 187 | 52,2 (45,6) | 44,0 | 137,5 (545,7) | 0,246 |
| ALT (UI/L) | 316 | 55,1 (91,4) | 149 | 61,4 (124,4) | 167 | 49,5 (44,9) | 1,000 | 252 | 53,1 (48,2) | 64,0 | 63,0 (180,2) | 0,023 |
| GGT (UI/L) | 243 | 101,7 (197,5) | 120 | 77,5 (70,0) | 123 | 125,4 (267,4) | 0,492 | 208 | 106,2 (212,4) | 35,0 | 75,1 (48,0) | 1,000 |
| Bilirubin (mg/dL) | 298 | 0,6 (0,5) | 141 | 0,6 (0,6) | 157 | 0,6 (0,4) | 0,584 | 242 | 0,6 (0,5) | 56,0 | 0,5 (0,3) | 0,840 |
| LDH (U/L) | 268 | 326,5 (165,3) | 132 | 283,2 (157,5) | 136 | 368,5 (162,3) | <0,001 | 216 | 310,7 (134,5) | 52,0 | 392,1 (247,8) | 0,006 |
| CRP (mg/dL) | 309 | 11,6 (10,7) | 144 | 7,7 (6,5) | 165 | 15,0 (12,4) | <0,001 | 241 | 10,4 (10,1) | 68,0 | 16,1 (11,9) | 0,001 |
| Ferritin (µg/L) | 201 | 850,3 (1317,4) | 99 | 550,0 (531,9) | 102 | 1141,7 (1728,6) | 0,014 | 171 | 828,0 (1258,1) | 30,0 | 977,5 (1634,3) | 0,840 |
| Procalcitonin (ng/mL) | 165 | 0,4 (0,8) | 64 | 0,3 (0,7) | 101 | 0,5 (0,9) | 0,020 | 135 | 0,3 (0,7) | 30,0 | 0,7 (1,1) | 0,002 |
| Lactate (mmol/L) | 65 | 1,8 (1,2) | 27 | 1,7 (0,9) | 38 | 1,8 (1,4) | 1,000 | 45 | 1,6 (0,8) | 20,0 | 2,1 (1,8) | 0,215 |
| $P_aO_2$ (mmHg) | 219 | 75,1 (28,6) | 90 | 79,3 (28,5) | 129 | 72,2 (28,5) | 0,134 | 169 | 75,8 (25,8) | 50,0 | 73,1 (36,9) | 0,316 |
| $P_aCO_2$ (mmHg) | 219 | 24 (3,2) | 90 | 24,2 (3,6) | 129 | 23,9 (2,8) | 0,915 | 169 | 24,1 (3,0) | 50,0 | 23,9 (3,6) | 0,786 |
| HCO3– (mmol/L) | 219 | 24,4 (2,5) | 90 | 24,5 (2,7) | 129 | 24,4 (2,3) | 1,000 | 169 | 24,5 (2,4) | 50,0 | 24,1 (2,9) | 0,368 |
| Ph | 219 | 7,5 (0,0) | 90 | 7,4 (0,0) | 129 | 7,4 (0,0) | 0,606 | 169 | 7,5 (0,0) | 50,0 | 7,4 (0,0) | 0,133 |

**ALT**: Aspartate-aminotransferasa. **AST**: Alanin-aminotransferase. **CPR**: C reactive protein. **GGT**: Gamma-glutamiltransferase. **INR**: international normalized ratio.

**LDH**: lactate dehydrogenase. **$P_aO_2$**: Partial pressure of oxygen. **$P_aCO_2$** Partial pressure of CO2.

marker of poor prognosis than arterial hypertension or diabetes mellitus. In our environment, Giacomelli et al. [10] also found that obesity was a risk factor (case fatality) in a cohort (n = 233) of patients from Italy. This finding is important given its prevalence in Europe both in the general population and in patients hospitalized with COVID-19 (20–25% and approximately 20%, respectively) [21]. In addition to the adverse mechanical effect on lung function (decrease in forced expiratory volume and forced vital capacity), it has been proposed that the metabolic alterations produced by COVID-19 could decrease cardiorespiratory reserves in the face of a stressor, enhance dysregulation of the immune system, and favor a prothrombotic and proinflammatory state, all of which are physiopathological phenomena relevant in SARS-CoV-2 infection [22].

Regarding previous pharmacological treatments, we believe that the increased risk associated with antipsychotics may be due to age and dementia (which in turn is related to limitation of therapeutic effort), rather than an intrinsic effect of these drugs. In our study, ACE inhibitors were not associated with a worse prognosis, which has also been found by other authors

[15–17]. We emphasize that in our sample, oral corticosteroids were predictors, rather than protectors, of death, which does not support the initial theories regarding their probable protective role. The Recovery clinical trial has recently showed that treatment with low dose dexamethasone decreases mortality in COVID-19 patients [23]. We have analyzed the prognostic role of corticosteroids, when used before the onset of COVID-19 disease, not as a treatment for it; therefore, we suggest that corticosteroids do not have a preventive role. Possibly corticosteroids are useful at certain stages of the disease, when inflammation is present, as the RECOVERY trial researchers suggest in the publication of the results.

Regarding disease symptoms, notably, dyspnea was a marker of severe disease but not an independent predictor of death. This could be related to the proposed hypothesis of "silent" hypoxia as a clinical manifestation in some affected patients [24]. On the other hand, in our sample, the great predictive capacity of cough (as a protector) with respect to death stands out. Our results refute those of other studies in which it was found that cough was an adverse predictor of case fatality or severe disease [25, 26]; all of these studies involved exclusively Asian cohorts. Additionally, fewer patients died who presented other nonrespiratory symptoms (diarrhea, arthromyalgia, headache, and alterations in smell and taste). However, regarding this result, we must recognize the possible existence of an information bias because the absence of dyspnea (poor prognostic factor) could have led clinicians to investigate other symptoms; therefore, these symptoms would have been collected with more frequency in patients without dyspnea, who have a better prognosis. Mental confusion, as a presenting symptom, was a predictor of case fatality in our sample, which we believe is due to its relationship with age.

The strong predictive capacity of the parameters related to respiratory involvement (oxygen saturation and number of observed radiological quadrants) and the inflammatory state (CRP in the emergency room) coincides with that reported in other studies [27] that highlight the prognostic importance of these factors. In addition, our study showed a shorter time of evolution of symptoms to emergency care in the group of patients who died (almost two days), with respect to the survivors. This suggests that a longer presentation may be a reflection of less aggressive disease, which is an interesting observation.

Regarding laboratory parameters upon admission, it is not surprising that CRP was the most powerful predictor of severe disease given the role of inflammation in the disease. However, it is interesting to note that inflammatory parameters were not independent predictors of case fatality in our sample. This finding, which contrasts with previous studies, it is possibly due to the different profile of the Spanish population with respect to the Asian one [6, 7]; the Spanish population has a greater burden of comorbidity, which may play an important role in mortality associated to COVID-19.

The protective role of eosinophilia, independent of other laboratory parameters, has not been evaluated or reported in previous studies. As eosinophilia was measured as a percentage of eosinophils with respect to the total, it could also reflect a decrease in another cell series (for example, neutrophils). If the protective role of eosinophilia is confirmed in other studies, this finding may have practical utility, if considered in prognostic scales, in addition to contributing to future knowledge on immune system reactions against SARS-CoV-2.

Our study was carried out on a hospitalized sample, so its results may not be applicable to patients with milder disease, who did not require hospitalization. Notably, our results involve a cohort from secondary hospitals (intermediate complexity) and a specific geographical area, which limits the generalization of the results to other cohorts, especially those of patients hospitalized in tertiary hospital centers (maximum complexity). Although we have an intensive care unit that doubled its capacity at the peak of the epidemic, it is likely that some of the most severe patients were transferred to tertiary hospitals and therefore remained underrepresented in our cohort.

Another limitation that should be mentioned is possible information bias because data extracted from clinical histories were used; these data were collected to guarantee the clinical care of the patients and not for the purpose of this research. This can affect the recording of extrapulmonary symptom presentation, as previously discussed. However, given that the majority of variables recorded are routinely used in clinical practice and are recorded reliably, for the best care of patients, we assume that if there was an information bias, this was limited or of little impact on the analyses.

In summary, advanced age, male sex and obesity were the main markers of poor prognosis in patients with COVID-19. The most frequent presenting symptom was fever; dyspnea was associated with severe disease, and the presence of cough was associated with greater survival. Low oxygen saturation in the emergency room, elevated CRP in the emergency room and initial radiological involvement were all related to worse prognosis.

## Acknowledgments

This research was conducted by the **COVID-19 research group of CSAPG** (led by Alejandro Rodríguez-Molinero: e-mail: rodriguez.molinero@gmail.com), which includes, in addition to the authors of this papers: Alberti Casas, Anna PhD, MD; Avalos Garcia, Jose L MD; Borrego Ruiz, Manel BS; Añaños Carrasco, Gemma MD; Campo Pisa, Pedro L; Capielo Fornerino, Ana M. MD; Chamero Pastilla, Antonio MD; Collado, Isabel MD; Fenollosa Artés, Andreu MD; Gris Ambros, Clara MD; Hernandez Martinez, Lourdes MD; Martín Puig, Mireia MD; Molina Hinojosa, José C. MD; Peramiquel Fonollosa, Laura MD; Pisani Zambrano, Italo G. MD; Rives, Juan P. MD; Sabria Bach, Enric. MD; Sanchez Rodriguez, Yris M. MD; Segura Martin, Maria del Mar. RN; Tremosa Llurba, Gemma MD; Ventosa Gili, Ester MD; Venturini Cabanellas, Florencia I. MD; Vidal Meler, Natalia. MD. Group affiliations: Àrea de Recerca, Consorci Sanitari de l'Alt Penedès i Garraf, Vilafranca del Penedès, Barcelona (Spain).

We would like to thank Gloria Moes for her invaluable help in coordinating the fieldwork. Gloria Alba, Nuria Pola and Anna María Soler, for their initial help in collecting drug data. Montserrat Pérez and Rosa Guilera, for their help with the electronic medical record, and to David Blancas and Lourdes Gabarró for their work in the hospital protocols for COVID-19, and their initial supply of bibliography. We also should thank the CSAPG informatics team, for their support during the study. Finally, we should thank to the manager of the Consorci Sanitari de l'Alt Penedès i Garraf, José Luis Ibáñez Pardos, and the management team, for making this study possible.

## Author Contributions

**Conceptualization:** Alejandro Rodríguez-Molinero, César Gálvez-Barrón.

**Data curation:** Oscar Macho, Gabriela F. López, Maria Teresa Robles, María Dolores Dapena, Sergi Martínez, Núria Milà Ràfols, Ernesto E. Monaco, Antonio Hidalgo García.

**Formal analysis:** Antonio Miñarro.

**Investigation:** Alejandro Rodríguez-Molinero, César Gálvez-Barrón.

**Methodology:** Alejandro Rodríguez-Molinero, César Gálvez-Barrón.

**Project administration:** Alejandro Rodríguez-Molinero.

**Supervision:** Alejandro Rodríguez-Molinero.

**Writing – original draft:** Alejandro Rodríguez-Molinero.

**Writing – review & editing:** César Gálvez-Barrón, Antonio Miñarro, Oscar Macho, Gabriela
F. López, Maria Teresa Robles, María Dolores Dapena, Sergi Martínez, Núria Milà Ràfols,
Ernesto E. Monaco, Antonio Hidalgo García.

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
