## [Decision Letter · Decision Letter 0]

14 Aug 2020

PONE-D-20-22605

Association between COVID-19 prognosis and disease presentation, comorbidities and chronic treatment of hospitalized patients.

PLOS ONE

Dear Dr. Rodríguez-Molinero,

Thank you for submitting your manuscript to PLOS ONE. After careful consideration, we feel that it has merit but does not fully meet PLOS ONE’s publication criteria as it currently stands. Therefore, we invite you to submit a revised version of the manuscript that addresses the points raised during the review process.

We look forward to receiving your revised manuscript.

Kind regards,

Wenbin Tan, Ph.D.

Academic Editor

PLOS ONE

Review Comments to the Author

Reviewer #1: It is important to present the risk factors for bad outcome in patients with covid-19 from different populations, even if most results like in the actual study are confirming the previous reports.

The analyses are well done and clearly presented.

I have only minor comments:

In Introduction, row 3, it is written that mortality is 4%, I suppose the authors mean case fatality and need to specify if it is in admitted patients. Mortality is the proportion of deaths in a population and for outcome of admitted patients the term case fatality is normally used for the proportion with fatal outcome. The mortality due to covid 19 is not as high as 4 % in any general population I am aware of.

I suggest that mortality should be replaced by case fatality also at

page 14, comorbidities, second section, row 6 and 10

page 15, row 6 and 9

in table 2

page 20 row 3,10,21

Decimals should be marked with dot (.) in table 1 and 2

Reviewer #2: Rodriguez Molinero et al present data on risk factors for severe disease and mortality in hospitalised patients with covid 19. The manuscript is consice and well written. The risk factors identified have been described previously but I believe this is good work that confirms previous reports.

My major concern is how the patient sample presented relates to the total number of cases of covid19 in the catchment area (including patients that were not hospitalised) during the study period. A short section describing this would be of value.

Secondly, as the authors acknowledge there is a risk of false discovery due to multiple comparisons than has been adjusted for. However, in multivariate models it is important to analyse whether different variables are interconnected. A section describing how this was analysed and adjusted for would be of value.

Minor comment:

The observation that eosinophilia was associated with better prognosis/less respiratory support can be presented and discussed but I do not think this should be presented in the conclusion. As stated multiple comparisons introduce a risk of false discoveries and unexpected findings should be interpreted with caution. I recommend that the authors remove this from the conclusion section of the abstract and the discussion.

---

## [Author Response · Author response to Decision Letter 0]

24 Aug 2020

We would like to thank to Dr. Wenbin Tan for the opportunity of reviewing the manuscript. 

RESPONSE TO THE REVIEWERS

Reviewer #1: It is important to present the risk factors for bad outcome in patients with covid-19 from different populations, even if most results like in the actual study are confirming the previous reports.The analyses are well done and clearly presented.

1.- I have only minor comments: In Introduction, row 3, it is written that mortality is 4%, I suppose the authors mean case fatality and need to specify if it is in admitted patients. Mortality is the proportion of deaths in a population and for outcome of admitted patients the term case fatality is normally used for the proportion with fatal outcome. The mortality due to covid 19 is not as high as 4 % in any general population I am aware of.

RESPONSE: Thanks for the reviewer’s observation. The reviewer is right, we should have used the term case fatality, instead of mortality, so we have corrected it and updated the data according to the last available report by WHO. The correspondent reference has also been updated.

2.- I suggest that mortality should be replaced by case fatality also at

page 14, comorbidities, second section, row 6 and 10

page 15, row 6 and 9 in table 2 page 20 row 3,10,21

RESPONSE: We have now reviewed the full text according to the reviewer’s recommendation (including all lines highlighted by the reviewer).

3.- Decimals should be marked with dot (.) in table 1 and 2

RESPONSE: We have corrected this too. 

Thanks very much for the review and the positive comments. 

Reviewer #2: Rodriguez Molinero et al present data on risk factors for severe disease and mortality in hospitalised patients with covid 19. The manuscript is consice and well written. The risk factors identified have been described previously but I believe this is good work that confirms previous reports.

1.- My major concern is how the patient sample presented relates to the total number of cases of covid19 in the catchment area (including patients that were not hospitalised) during the study period. A short section describing this would be of value.

RESPONSE: The total number of cases of Covid-19 in the catchment area were unknown at the time of the study. The epidemic was at its worst moment and PCR was only performed on severe patients. Milder cases were sent home from the primary care settings or from the emergency room without PCR investigation, and there were warnings for the population not to go to the emergency services or health centers, if they had mild symptoms (they should stay at home). For all these reasons the incidence of the disease at that time is not calculable. 

However, our hospitals are the only hospitals in their reference area, so they must have brought together most of the cases that required admission and therefore, our sample possibly represents well the population with COVID-19 that requires hospitalization in our area.

Following the reviewer's comment, we did some research to see if we could get COVID-19 numbers in the community at the time of the study. We have found that, during the study period, 1442 people were diagnosed with COVID-19 by nasal PCR, in our geographic area (including hospitalized and community patients). Of them, a significant proportion (418 patients) have been included in our sample.

Now we have added the total number of PCR diagnosed COVID-19 in our area to the text (see methods section first paragraph) , and explained the problems to generalize the results to milder patients in the discussion (see 7th paragraph of the discussion section)

2.- Secondly, as the authors acknowledge there is a risk of false discovery due to multiple comparisons than has been adjusted for. However, in multivariate models it is important to analyse whether different variables are interconnected. A section describing how this was analysed and adjusted for would be of value.

RESPONSE: We have understood that the reviewer refers to the possibility of multicollinearity or excess correlation between the variables of the model. There are various strategies to identify or correct the problems of multicoliniality. One of them is the use of penalized regression models, such as the LASSO method that we have used in our models.

We have added a sentence to the text indicating that we have treated possible multicollinearity problems with the LASSO method, and also, we have added a bibliographic reference that justifies the use of the technique for this purpose.

Please see changes in the third to last paragraph of the methods section. 

Minor comment:

3.- The observation that eosinophilia was associated with better prognosis/less respiratory support can be presented and discussed but I do not think this should be presented in the conclusion. As stated multiple comparisons introduce a risk of false discoveries and unexpected findings should be interpreted with caution. I recommend that the authors remove this from the conclusion section of the abstract and the discussion.

RESPONSE: We consider that the reviewer is right. We have withdrawn this finding from the conclusion. Furthermore, the variable measures the number of eosinophils in relative terms (percentage in proportion to total leukocytes), therefore, it may actually be reflecting a cytopenia from another series. We have clarified this fact briefly in the discussion, and removed the finding from the conclussion.

Please see changes in results “laboratory analytical parameters”, 2nd paragraph, Discussion section, 6th paragraph, and conclusions. 

Thanks for your careful review.

---

## [Editor Report · Decision Letter 1]

10 Sep 2020

Association between COVID-19 prognosis and disease presentation, comorbidities and chronic treatment of hospitalized patients.

PONE-D-20-22605R1

Dear Dr. Rodríguez-Molinero,

We’re pleased to inform you that your manuscript has been judged scientifically suitable for publication and will be formally accepted for publication once it meets all outstanding technical requirements.

Kind regards,

Wenbin Tan

Academic Editor

PLOS ONE

Additional Editor Comments:

The authors have responded all comments well and thoroughly.
---

## [Editor Report · Acceptance letter]

1 Oct 2020

PONE-D-20-22605R1 

Association between COVID-19 prognosis and disease presentation, comorbidities and chronic treatment of hospitalized patients. 

Dear Dr. Rodríguez-Molinero:

I'm pleased to inform you that your manuscript has been deemed suitable for publication in PLOS ONE. Congratulations! Your manuscript is now with our production department. 

Kind regards, 

on behalf of

Dr. Wenbin Tan 

Academic Editor

PLOS ONE